# MULTI-LABEL LEARNING WITH THE RNNS FOR FASHION SEARCH

**Se-Yeoung Kim, Sang-Il Na, Ha-Yoon Kim, Moon-Ki Kim, Byoung-Ki Jeon**
Machine Intelligence Lab., SK Planet
Seongnam City, South Korea
{seyeong,sang.il.na,hayoon,moonki,standard}@sk.com

**Taewan Kim** *
Naver Labs, Naver Corp.
Seongnam City, South Korea
{taey.16@navercorp.com}

## ABSTRACT

We build a large-scale visual search system which finds similar product images given a fashion item. Defining similarity among arbitrary fashion-products is still remains a challenging problem, even there is no exact ground-truth. To resolve this problem, we define more than 90 fashion-related attributes, and combination of these attributes can represent thousands of unique fashion-styles. We then introduce to use the recurrent neural networks (RNNs) recognising multiple fashion-attributes with the end-to-end manner. To build our system at scale, these fashion-attributes are again used to build an inverted indexing scheme. In addition to these fashion-attributes for semantic similarity, we extract colour and appearance features in a region-of-interest (ROI) of a fashion item for visual similarity. By sharing our approach, we expect active discussion on that how to apply current deep learning researches into the e-commerce industry.

## 1 INTRODUCTION

Online commerce has been a great impact on our life over the past decade. We focus on an online market for fashion related items[1]. Finding similar fashion-product images for a given image query is a classical problem in an application to computer vision, however, still challenging due to the absence of an absolute definition of the similarity between arbitrary fashion items.

Deep learning technology has given great success in computer vision tasks such as efficient feature representation (Razavian et al., 2014; Babenko et al., 2014), classification (He et al., 2016a; Szegedy et al., 2016b), detection (Ren et al., 2015; Zhang et al., 2016), and segmentation (Long et al., 2015). Furthermore, image to caption generation (Vinyals et al., 2015; Xu et al., 2015) and visual question answering (VQA) (Antol et al., 2015) are emerging research fields combining vision, language (Mikolov et al., 2010), sequence to sequence (Sutskever et al., 2014), long-term memory (Xiong et al., 2016) based modelling technologies.

These computer vision researches mainly concern about general object recognition. However, in our fashion-product search domain, we need to build a very specialised model which can mimic human's perception of fashion-product similarity. To this end, we start by brainstorming about what makes two fashion items are similar or dissimilar. Fashion-specialist and merchandisers are also involved. We then compose fashion-attribute dataset for our fashion-product images. Table 1 explains a part of our fashion-attributes. Conventionally, each of the columns in Table 1 can be modelled as a multi-class classification. Therefore, our fashion-attributes naturally is modelled as a multi-label classification.

---

*This work was done by the author at SK Planet.

[1]In our e-commerce platform, 11st (http://english.11st.co.kr/html/en/main.html), almost a half of user-queries are related to the fashion styles, and clothes.

Table 1: An example of fashion-attributes.

| Great-category (3 classes) | Fashion-category (19 classes) | Gender (2 classes) | Silhouette (14 classes) | Collar (18 classes) | sleeve-length (6 classes) | ... |
|---|---|---|---|---|---|---|
| bottom | T-shirts | male | normal | shirt | long | ⋮ |
| top | pants | female | A-line | turtle | a half | ⋮ |
| ⋮ | bags | | ⋮ | round | sleeveless | ⋮ |
| | ⋮ | | ⋮ | ⋮ | ⋮ | ⋮ |

Multi-label classification has a long history in the machine learning field. To address this problem, a straightforward idea is to split such multi-labels into a set of multi-class classification problems. In our fashion-attributes, there are more than 90 attributes. Consequently, we need to build more than 90 classifiers for each attribute. It is worth noting that, for example, *collar* attribute can represent the upper-garments, but it is absent to represent bottom-garments such as skirts or pants, which means some attributes are conditioned on other attributes. This is the reason that the learning tree structure of the attributes dependency can be more efficient (Zhang & Zhang, 2010; Fu et al., 2012; Gibaja & Ventura, 2015).

Recently, recurrent neural networks (RNN) are very commonly used in automatic speech recognition (ASR) (Graves et al., 2013; Graves & Jaitly, 2014), language modelling (Mikolov et al., 2010), word dependency parsing (Mirowski & Vlachos, 2015), machine translation (Cho et al., 2014), and dialog modelling (Henderson et al., 2014; Serban et al., 2016). To preserve long-term dependency in hidden context, Long-Short Term Memory (LSTM) (Hochreiter & Schmidhuber, 1997) and its variants (Zaremba et al., 2014; Cooijmans et al., 2016) are breakthroughs in such fields. We use this LSTM to learn fashion-attribute dependency structure *implicitly*. By using the LSTM, our attribute recognition problem is regarded to as a sequence classification. There is a similar work in Wang et al. (2016), however, we do not use the VGG16 network (Simonyan & Zisserman, 2014) as an image encoder but use our own encoder. To the best of our knowledge, it is the first work applying LSTM into a multi-label classification task in the commercial fashion-product search domain.

The remaining of this paper is organized as follows. In Sec. 2, We describe details about our fashion-attribute dataset. Sec. 3 describes the proposed fashion-product search system in detail. Sec. 4 explains empirical results given image queries. Finally, we draw our conclusion in Sec. 5.

## 2 Building the fashion-attribute dataset

We start by building large-scale fashion-attribute dataset in the last year. We employ maximum 100 man-months and take almost one year for completion. There are 19 fashion-categories and more than 90 attributes for representing a specific fashion-style. For example, *top* garments have the *T-shirts*, *blouse*, *bag* etc. The *T-shirts* category has the *collar*, *sleeve-length*, *gender*, etc. The *gender* attribute has binary classes (i.e. female and male). *Sleeve-length* attribute has multiple classes (i.e. long, a half, sleeveless etc.). Theoretically, the combination of our attributes can represent thousands of unique fashion-styles. A part of our attributes are in Table 1. ROIs for each fashion item in an image are also included in this dataset. Finally, we collect 1 million images in total. This internal dataset is to be used for training our fashion-attribute recognition model and fashion-product ROI detector respectively.

## 3 Fashion-product search system

In this section, we describe the details of our system. The whole pipeline is illustrated in Fig. 3. As a conventional information retrieval system, our system has offline and online phase. In offline process, we take both an image and its textual meta-information as the inputs. The reason we take additional textual meta-information is that, for example, in Fig. 1a dominant fashion item in the image is a white dress however, our merchandiser enrolled it to sell the brown cardigan as described

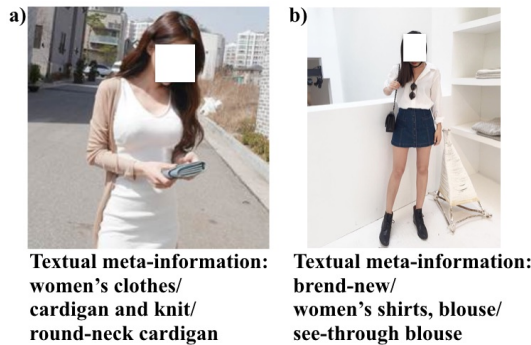

a) Textual meta-information:
women's clothes/
cardigan and knit/
round-neck cardigan

b) Textual meta-information:
brend-new/
women's shirts, blouse/
see-through blouse

Figure 1: Examples of image and its textual meta-information.

in its meta-information. In Fig. 1b, there is no way of finding which fashion item is to be sold without referring the textual meta-information seller typed manually. Therefore, knowing intension (i.e. what to sell) for our merchandisers is very important in practice. To catch up with these intension, we extract *fashion-category information* from the textual meta. The extracted *fashion-category information* is fed to the fashion-attribute recognition model. The fashion-attribute recognition model predicts a set of fashion-attributes for the given image. (see Fig. 2) These fashion-attributes are used as keys in the inverted indexing scheme. On the next stage, our fashion-product ROI detector finds where the fashion-category item is in the image. (see Fig. 8) We extract colour and appearance features for the detected ROI. These visual features are stored in a postings list. In these processes, it is worth noting that, as shown in Fig. 8, our system can generate different results in the fashion-attribute recognition and the ROI detection for the same image by *guiding the fashion-category information*. In online process, there is two options for processing a user-query. We can

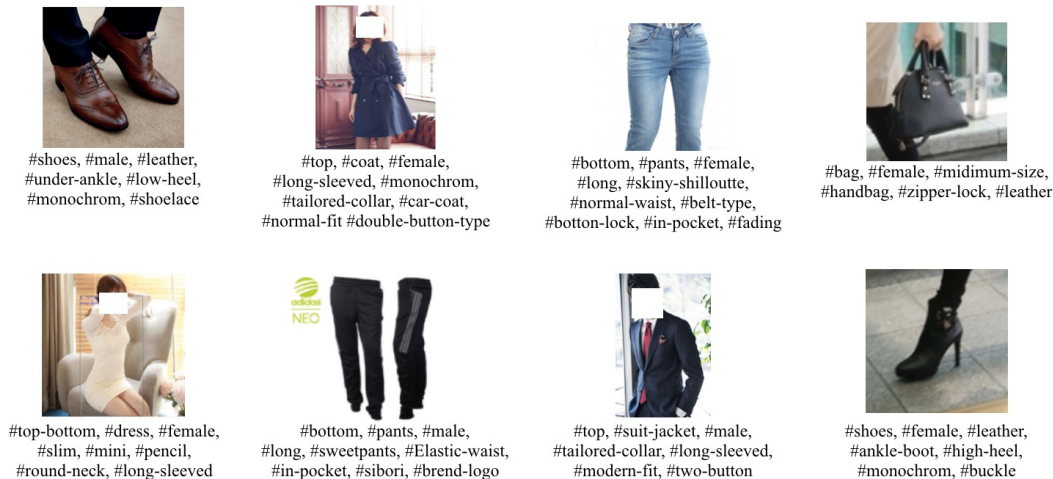

#shoes, #male, #leather,
#under-ankle, #low-heel,
#monochrom, #shoelace

#top, #coat, #female,
#long-sleeved, #monochrom,
#tailored-collar, #car-coat,
#normal-fit #double-button-type

#bottom, #pants, #female,
#long, #skiny-shilloutte,
#normal-waist, #belt-type,
#botton-lock, #in-pocket, #fading

#bag, #female, #midimum-size,
#handbag, #zipper-lock, #leather

#top-bottom, #dress, #female,
#slim, #mini, #pencil,
#round-neck, #long-sleeved

#bottom, #pants, #male,
#long, #sweetpants, #Elastic-waist,
#in-pocket, #sibori, #brend-logo

#top, #suit-jacket, #male,
#tailored-collar, #long-sleeved,
#modern-fit, #two-button

#shoes, #female, #leather,
#ankle-boot, #high-heel,
#monochrom, #buckle

Figure 2: Examples of recognized fashion-attributes for given images.

take a *guided* information, what the user wants to find, or the fashion-attribute recognition model automatically finds what fashion-category item is the most likely to be queried. This is up to the user's choice. For the given image by the user, the fashion-attribute recognition model generates fashion-attributes, and the results are fed into the fashion-product ROI detector. We extract colour and appearance features in the ROI resulting from the detector. We access to the inverted index addressed by the generated set of fashion-attributes, and then get a postings list for each fashion-attribute. We perform nearest-neighbor retrieval in the postings lists so that the search complexity is reduced drastically while preserving the semantic similarity. To reduce memory capacity and speed up this nearest-neighbor retrieval process once more, our features are binarized and CPU depen-

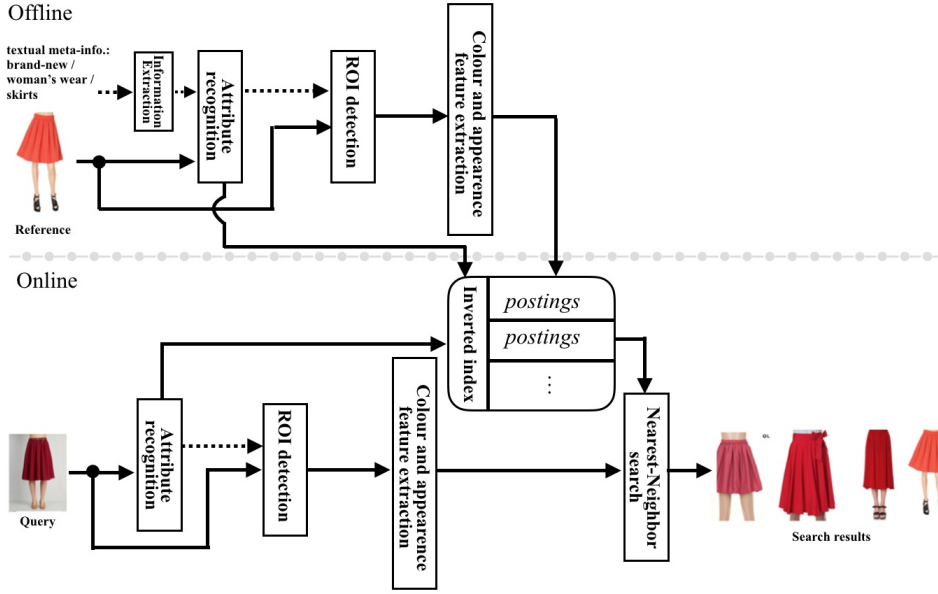

Figure 3: The whole pipeline of the proposed fashion-product search system. (Dashed lines denote the flows of the *guided* information.)

dent intrinsic instruction (i.e. assembly `popcnt` instruction[2]) is used for computing the hamming distance.

## 3.1 VISION ENCODER NETWORK

We build our own vision encoder network (ResCeption) which is based on inception-v3 architecture (Szegedy et al., 2016b). To improve both speed of convergence and generalization, we introduce a shortcut path (He et al., 2016a;b) for each data-flow stream (except streams containing one convolutional layer at most) in all inception-v3 modules. Denote input of $l$-th layer , $\mathbf{x}^l \in \mathbb{R}$ , output of the $l$-th layer, $\mathbf{x}^{l+1}$, a $l$-th layer is a function, $\mathcal{H} : \mathbf{x}^l \mapsto \mathbf{x}^{l+1}$ and a loss function, $\mathcal{L}(\theta; \mathbf{x}^L)$. Then forward and back(ward)propagation is derived such that

$$\mathbf{x}^{l+1} \quad = \quad \mathcal{H}(\mathbf{x}^l) + \mathbf{x}^l \tag{1}$$

$$\frac{\partial \mathbf{x}^{l+1}}{\partial \mathbf{x}^l} \quad = \quad \frac{\partial \mathcal{H}(\mathbf{x}^l)}{\partial \mathbf{x}^l} + \mathbf{1} \tag{2}$$

Imposing gradients from the loss function to $l$-th layer to Eq. (2),

$$\begin{aligned}
\frac{\partial \mathcal{L}}{\partial \mathbf{x}^l} \quad &:= \quad \frac{\partial \mathcal{L}}{\partial \mathbf{x}^L} \cdots \frac{\partial \mathbf{x}^{l+2}}{\partial \mathbf{x}^{l+1}} \frac{\partial \mathbf{x}^{l+1}}{\partial \mathbf{x}^l} \\
&= \quad \frac{\partial \mathcal{L}}{\partial \mathbf{x}^L} \Big( \mathbf{1} + \cdots + \frac{\partial \mathcal{H}(\mathbf{x}^{L-2})}{\partial \mathbf{x}^l} + \frac{\partial \mathcal{H}(\mathbf{x}^{L-1})}{\partial \mathbf{x}^l} \Big) \\
&= \quad \frac{\partial \mathcal{L}}{\partial \mathbf{x}^L} \Big( \mathbf{1} + \sum_{i=L-1}^{l} \frac{\partial \mathcal{H}(\mathbf{x}^i)}{\partial \mathbf{x}^l} \Big).
\end{aligned} \tag{3}$$

As in the Eq. (3), the error signal, $\frac{\partial \mathcal{L}}{\partial \mathbf{x}^L}$, goes down to the $l$-th layer *directly* through the shortcut path, and then the gradient signals from $(L-1)$-th layer to $l$-th layer are added consecutively (i.e. $\sum_{i=L-1}^{l} \frac{\partial \mathcal{H}(\mathbf{x}^i)}{\partial \mathbf{x}^l}$). Consequently, all of terms in Eq. (3) are aggregated by the additive operation instead of the multiplicative operation except initial error from the loss (i.e. $\frac{\partial \mathcal{L}}{\partial \mathbf{x}^L}$). It prevents from vanishing or exploding gradient problem. Fig. 4 depicts network architecture for shortcut

---

[2]http://www.gregbugaj.com/?tag=assembly (accessed at Aug. 2016)

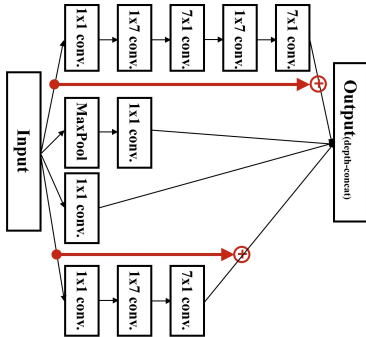

Figure 4: Network architecture for shortcut paths (depicted in two red lines) in an inception-v3 module.

paths in an inception-v3 module. We use projection shortcuts throughout the original inception-v3 modules due to the dimension constraint.[3] To demonstrate the effectiveness of the shortcut paths in the inception modules, we reproduce ILSVRC2012 classification benchmark (Russakovsky et al., 2015) for inception-v3 and our ResCeption network. As in Fig. 5a, we verify that residual shortcut paths are beneficial for fast training and slight better generalization.[4] The whole of the training curve is shown in Fig. 5b. The best validation error is reached at 23.37% and 6.17% at top-1 and top-5, respectively. That is a competitive result.[5] To demonstrate the representation power of our ResCeption, we employ the transfer learning strategy for applying the pre-trained ResCeption as an image encoder to generate captions. In this experiment, we verify our ResCeption encoder outperforms the existing VGG16 network[6] on MS-COCO challenge benchmark (Chen et al., 2015). The best validation CIDEr-D score (Vedantam et al., 2015) for c5 is 0.923 (see Fig. 5c) and test CIDEr-D score for c40 is 0.937.[7]

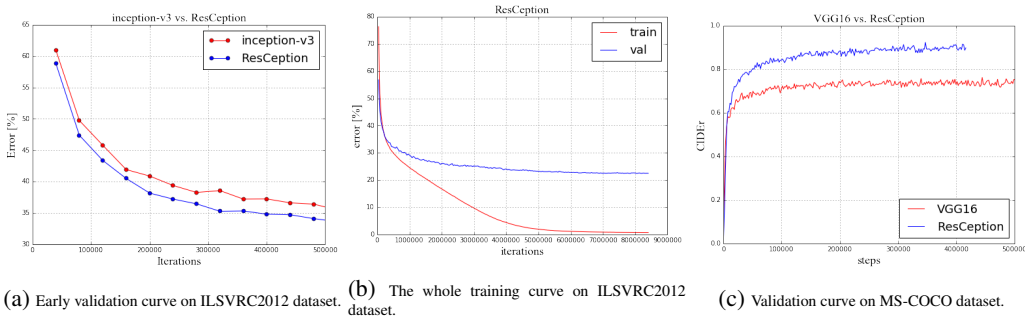

(a) Early validation curve on ILSVRC2012 dataset.  (b) The whole training curve on ILSVRC2012 dataset.  (c) Validation curve on MS-COCO dataset.

Figure 5: Training curves on ILSVRC2012 and MS-COCO dataset with our ResCeption model.

### 3.2 MULTI-LABEL LEARNING AS SEQUENCE PREDICTION BY USING THE RNN

The traditional multi-class classification associates an instance $x$ with a single label $a$ from previously defined a finite set of labels $A$. The multi-label classification task associates several finite sets of labels $A_n \subset \mathcal{A}$. The most well known method in the multi-label literature are the binary relevance method (BM) and the label combination method (CM). There are drawbacks in both BM

---

[3]If the input and output dimension of the main-branch is not the same, projection shortcut should be used instead of identity shortcut.

[4]This is almost the same finding from Szegedy et al. (2016a) but our work was done independently.

[5]http://image-net.org/challenges/LSVRC/2015/results

[6]https://github.com/torch/torch7/wiki/ModelZoo

[7]We submitted our final result with beam search on MS-COCO evaluation server and found out the beam search improves final CIDEr-D for c40 score by 0.02.

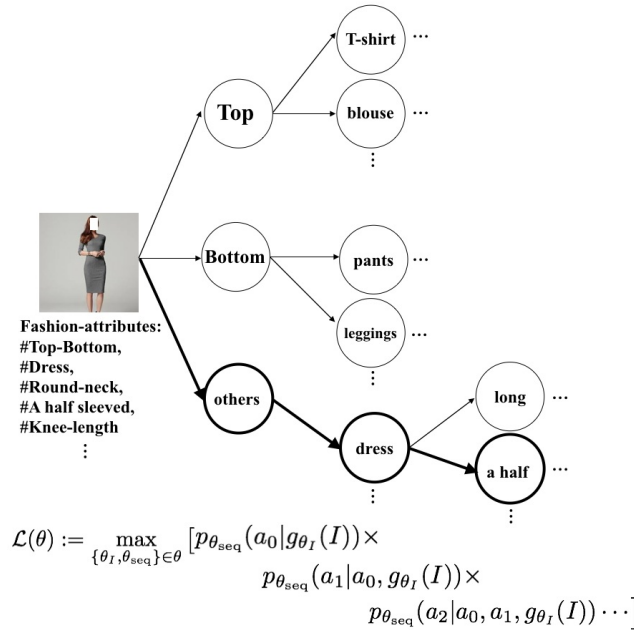

$$\mathcal{L}(\theta) := \max_{\{\theta_I, \theta_{\text{seq}}\} \in \theta} \left[ p_{\theta_{\text{seq}}}(a_0 | g_{\theta_I}(I)) \times \right.$$
$$p_{\theta_{\text{seq}}}(a_1 | a_0, g_{\theta_I}(I)) \times$$
$$\left. p_{\theta_{\text{seq}}}(a_2 | a_0, a_1, g_{\theta_I}(I)) \cdots \right]$$

Figure 6: An example of the fashion-attribute dependence tree for a given image and the objective function of our fashion-attribute recognition model.

and CM. The BM ignores label correlations that exist in the training data. The CM directly takes into account label correlations, however, a disadvantage is its worst-case time complexity (Read et al., 2009). To tackle these drawbacks, we introduce to use the RNN. Suppose we have random variables $a \in A_n, A_n \subset \mathcal{A}$. The objective of the RNN is to maximise the joint probability, $p(a_t, a_{t-1}, a_{t-2}, \ldots a_0)$, where $t$ is a sequence (time) index. This joint probability is factorized as a product of conditional probabilities recursively,

$$p(a_t, a_{t-1}, \ldots a_0) = \underbrace{\underbrace{p(a_0)p(a_1|a_0)}_{p(a_0,a_1)} p(a_2|a_1, a_0) \cdots}_{p(a_0,a_1,a_2,\ldots)} \tag{4}$$
$$= p(a_0) \prod_{t=1}^{T} p(a_t | a_{t-1}, \ldots, a_0).$$

Following the Eq. 4, we can handle multi-label classification as sequence classification which is illustrated in Fig. 6. There are many label dependencies among our fashion-attributes. Direct modelling of such label dependencies in the training data using the RNN is our key idea. We use the ResCeption as a vision encoder $\theta_I$, LSTM and softmax regression as our sequence classifier $\theta_{\text{seq}}$, and negative log-likelihood (NLL) as the loss function. We backpropagage gradient signal from the sequence classifier to vision encoder.[8] Empirical results of our ResCeption-LSTM based attribute recognition are in Fig. 2. Many fashion-category dependent attributes such as *sweetpants, fading, zipper-lock, mini*, and *tailored-collar* are recognized quite well. Fashion-category independent attributes (e.g., *male, female*) are also recognizable. It is worth noting we do *not* model the fashion-attribute dependance tree *at all*. We demonstrate the RNN learns attribute dependency structure *implicitly*. We evaluate our attribute recognition model on the fashion-attribute dataset. We split this dataset into 721544, 40000, and 40000 images for training, validating, and testing. We employ the early-stopping strategy to preventing over-fitting using the validation set. We measure precision and recall for a set of ground-truth attributes and a set of predicted attributes for each image. The quantitative results are in Table 2.

---

[8]Our attribute recognition model is parameterized as $\theta = [\theta_I; \theta_{\text{seq}}]$. In our case, updating $\theta_I$ as well as $\theta_{\text{seq}}$ in the gradient descent step helps for much better performance.

Table 2: A quantitative evaluation of the ResCeption-LSTM based attribute recognition model.

| Measurement | Train | Validation | Test |
|---|---|---|---|
| Precision | 0.866 | 0.842 | 0.841 |
| Recall | 0.867 | 0.841 | 0.842 |
| NLL | 0.298 | 0.363 | 0.363 |

### 3.3 *Guided* ATTRIBUTE-SEQUENCE GENERATION

Our prediction model of the fashion-attribute recognition is based on the sequence generation process in the RNN (Graves, 2013). The attribute-sequence generation process is illustrated in Fig. 7. First, we predict a probability of the first attribute for a given internal representation of the image i.e. $p_{\theta_{seq}}(a_0|g_{\theta_I}(I))$, and then sample from the estimated probability of the attribute, $a_0 \sim p_{\theta_{seq}}(a_0|g_{\theta_I}(I))$. The sampled symbol is fed to as the next input to compute $p_{\theta_{seq}}(a_1|a_0, g_{\theta_I}(I))$. This sequential process is repeated recursively until a sampled result is reached at the special end-of-sequence (EOS) symbol. In case that we generate a set of attributes for a *guided fashion-category*, we do not sample from the previously estimated probability, but select the *guided fashion-category*, and then we feed into it as the next input deterministically. It is the key to considering for each seller's intention. Results for the *guided* attribute-sequence generation is shown in Fig. 8.

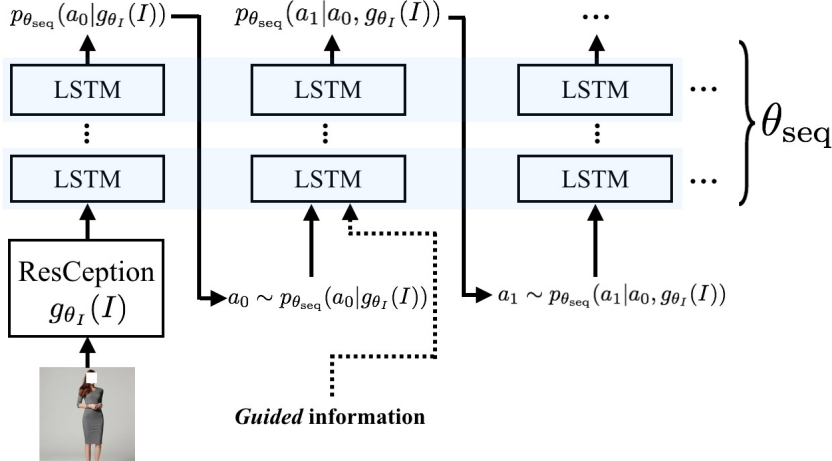

Figure 7: *Guided* sequence generation process.

### 3.4 *Guided* ROI DETECTION

Our fashion-product ROI detection is based on the Faster R-CNN (Ren et al., 2015). In the conventional multi-class Faster R-CNN detection pipeline, one takes an image and outputs a tuple of (ROI coordinate, object-class, class-score). In our ROI detection pipeline, we take additional information, *guided fashion-category* from the ResCeption-LSTM based attribute-sequence generator. Our fashion-product ROI detector finds where the *guided fashion-category* item is in a given image. Jing et al. (2015) also uses a similar idea, but they train several detectors for each category independently so that their works do not scale well. We train a detector for all fashion-categories jointly. Our detector produces ROIs for all of the fashion-categories at once. In post-processing, we reject ROIs that their object-classes are not matched to the *guided fashion-category*. We demonstrate that the *guided fashion-category* information contributes to higher performance in terms of mean average precision (mAP) on the fashion-attribute dataset. We measure the mAP for the intersection-of-union (IoU) between ground-truth ROIs and predicted ROIs. (see Table 3) That is due to the fact that our *guided fashion-category* information reduces the false positive rate. In our fashion-product search pipeline, the colour and appearance features are extracted in the detected ROIs.

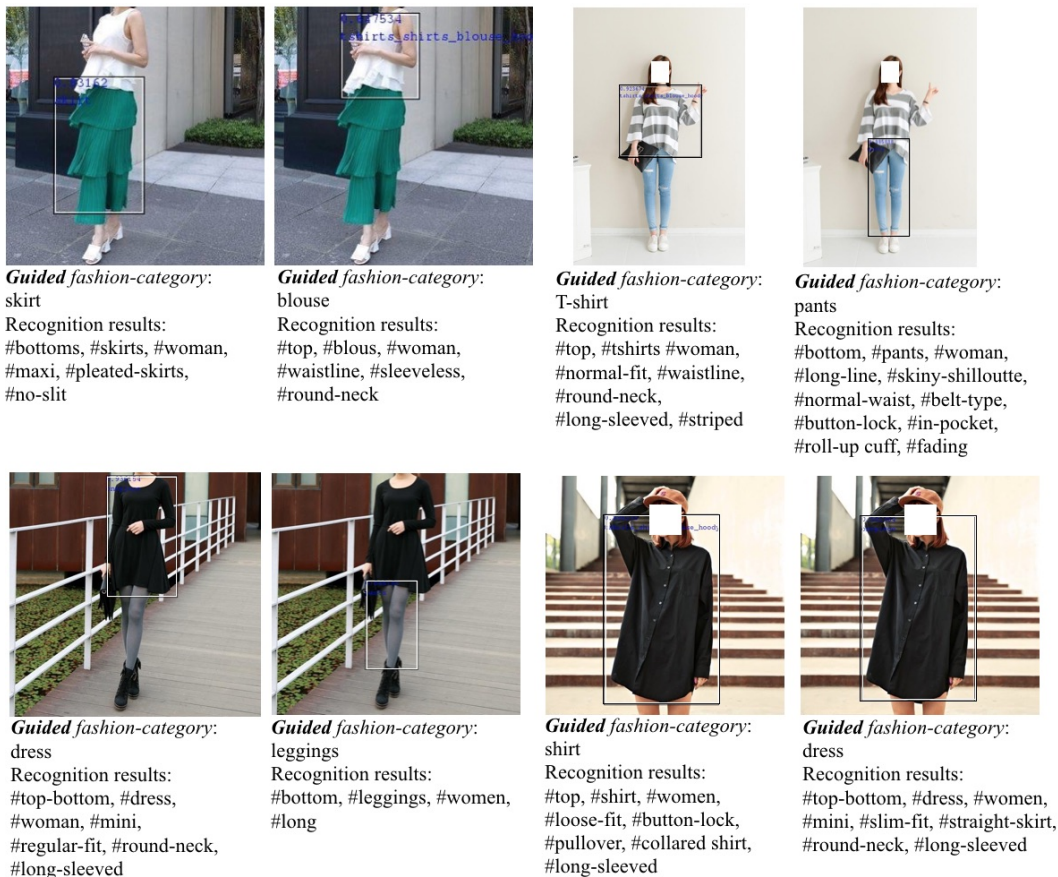

*Guided* fashion-category:
skirt
Recognition results:
#bottoms, #skirts, #woman,
#maxi, #pleated-skirts,
#no-slit

*Guided* fashion-category:
blouse
Recognition results:
#top, #blous, #woman,
#waistline, #sleeveless,
#round-neck

*Guided* fashion-category:
T-shirt
Recognition results:
#top, #tshirts #woman,
#normal-fit, #waistline,
#round-neck,
#long-sleeved, #striped

*Guided* fashion-category:
pants
Recognition results:
#bottom, #pants, #woman,
#long-line, #skiny-shilloutte,
#normal-waist, #belt-type,
#button-lock, #in-pocket,
#roll-up cuff, #fading

*Guided* fashion-category:
dress
Recognition results:
#top-bottom, #dress,
#woman, #mini,
#regular-fit, #round-neck,
#long-sleeved

*Guided* fashion-category:
leggings
Recognition results:
#bottom, #leggings, #women,
#long

*Guided* fashion-category:
shirt
Recognition results:
#top, #shirt, #women,
#loose-fit, #button-lock,
#pullover, #collared shirt,
#long-sleeved

*Guided* fashion-category:
dress
Recognition results:
#top-bottom, #dress, #women,
#mini, #slim-fit, #straight-skirt,
#round-neck, #long-sleeved

Figure 8: Examples of the consecutive process in the *guided* sequence generation and the *guided* ROI detection. Although we take the same input image, results can be totally different *guiding the fashion-category information*.

Table 3: Fashion-product ROI Detector evaluation. (mAP)

| IoU | 0.5 | 0.6 | 0.7 | 0.8 | 0.9 |
|---|---|---|---|---|---|
| Guided | **0.877** | **0.872** | **0.855** | **0.716** | **0.225** |
| Non-guided | 0.849 | 0.842 | 0.818 | 0.684 | 0.223 |

## 3.5 VISUAL FEATURE EXTRACTION

To extract appearance feature for a given ROI, we use pre-trained GoogleNet (Szegedy et al., 2015). In this network, both inception4 and inception5 layer's activation maps are used. We evaluate this feature on two similar image retrieval benchmarks, i.e. Holidays (Jegou et al., 2008) and UK-benchmark (UKB) (Nistér & Stewénius, 2006). In this experiment, we do not use any post-processing method or fine-tuning at all. The mAP on Holidays is 0.783, and the precision@4 and recall@4 on UKB is 0.907 and 0.908 respectively. These scores are competitive against several deep feature representation methods (Razavian et al., 2014; Babenko et al., 2014). Examples of queries and resulting nearest-neighbors are in Fig. 9. On the next step, we binarize this appearance feature by simply thresholding at 0. The reason we take this simple thresholding to generate the hash code is twofold. The neural activation feature map at a higher layer is a sparse and distributed code in nature. Furthermore, the bias term in a linear layer (e.g., convolutional layer) compensates for

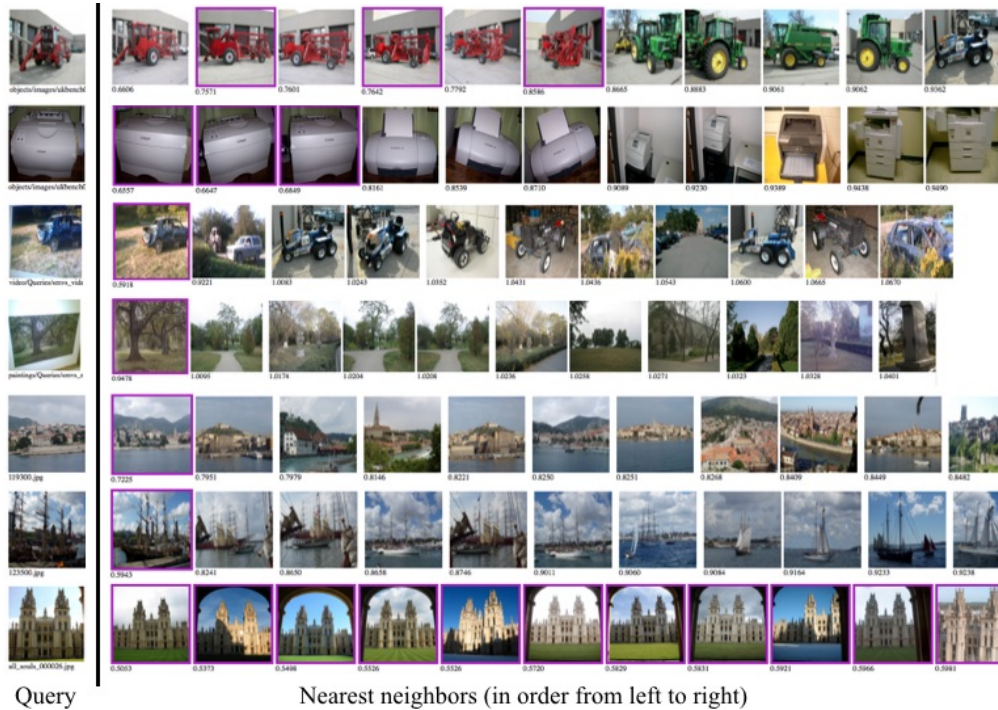

Figure 9: Examples of retrieved results on Holidays and UKB. The violet rectangles denote the ground-truth nearest-neighbors corresponding queries.

aligning zero-centering of the output feature space weakly. Therefore, we believe that a code from a well-trained neural model, itself, can be a good feature even to be binarized. In our experiment, such simple thresholding degrades mAP by 0.02 on the Holidays dataset, but this method makes it possible to scaling up in the retrieval. In addition to the appearance feature, we extract colour feature using the simple (bins) colour histogram in HSV space, and distance between a query and a reference image is computed by using the weighted combination of the two distances from the colour and the appearance feature.

## 4 EMPIRICAL RESULTS

To evaluate empirical results of the proposed fashion-product search system, we select 3 million fashion-product images in our e-commerce platform at random. These images are mutually exclusive to the fashion-attribute dataset. We have again selected images from the web used for the queries. All of the reference images pass through the offline process as described in Sec. 3, and resulting inverted indexing database is loaded into main-memory (RAM) by our daemon system. We send the pre-selected queries to the daemon system with the RESTful API. The daemon system then performs the online process and returns nearest-neighbor images correspond to the queries. In this scenario, there are three options to get similar fashion-product images. Option 1 is that the fashion-attribute recognition model automatically selects fashion-category, the most likely to be queried in the given image. Option 2 is that a user manually selects a fashion-category given a query image. (see Fig. 10) Option 3 is that a user draw a rectangle to be queried by hand like Jing et al. (2015). (see Fig. 11) By the recognized fashion-attributes, the retrieved results reflect the user's main needs, e.g. gender, season, utility as well as the fashion-style, that could be lacking when using visual feature representation only.

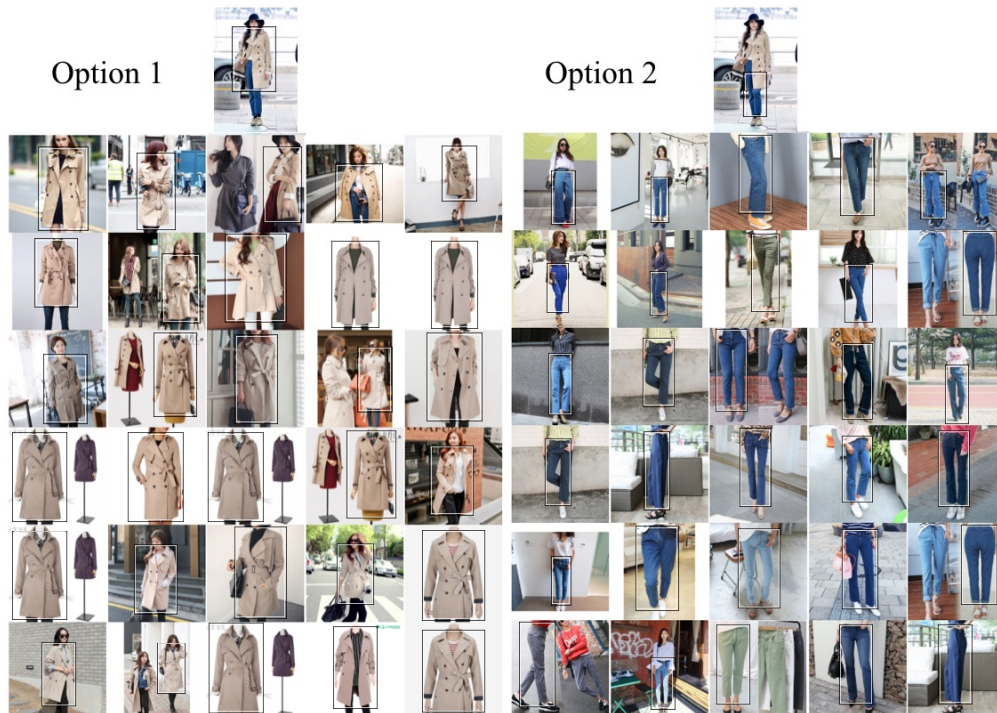

(a) For the Option2, the *guided* information is "pants".

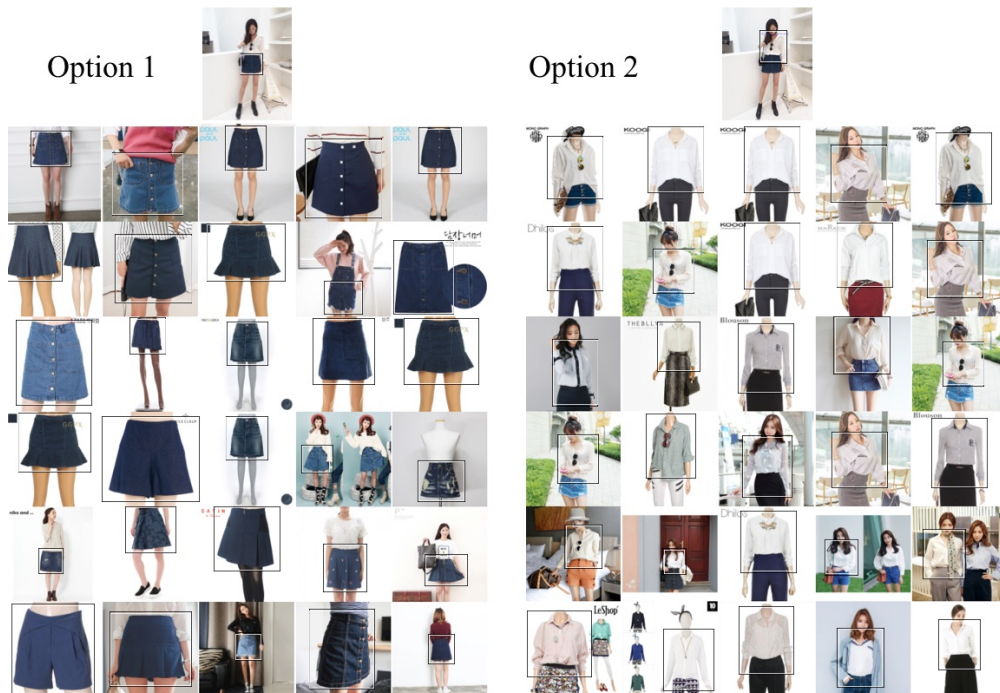

(b) For the option 2, the *guided* information is "blouse".

Figure 10: Similar fashion-product search for the Option 1 and the Option 2.

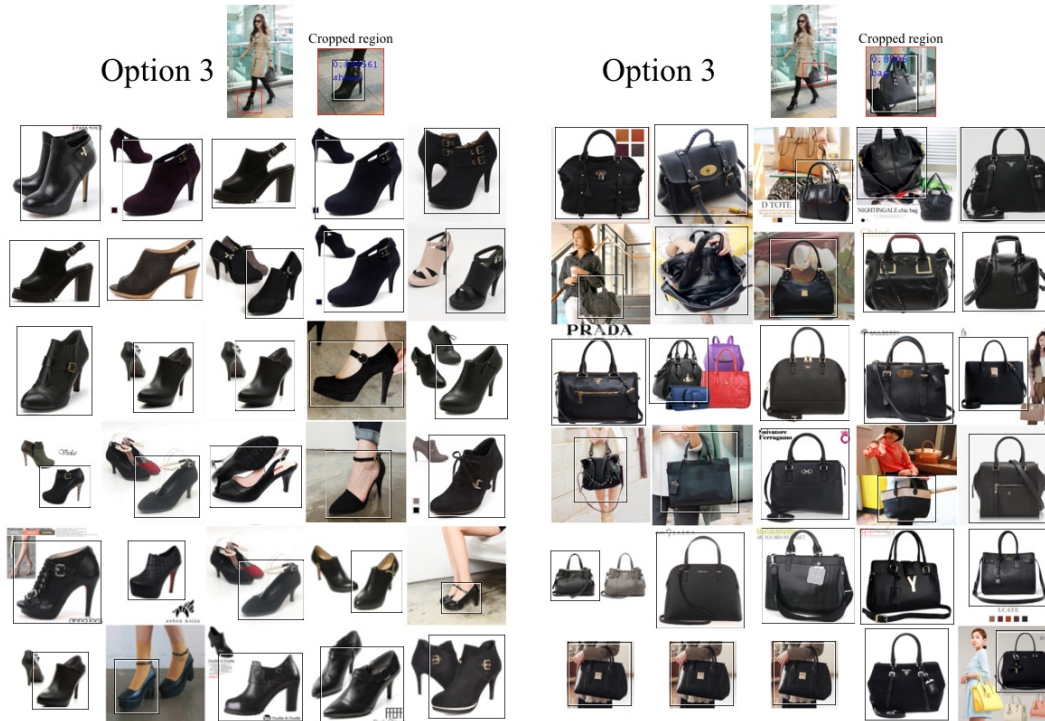

Figure 11: Similar fashion-product search for the Option 3.

## 5 CONCLUSIONS

Today's deep learning technology has given great impact on various research fields. Such a success story is about to be applied to many industries. Following this trend, we traced the start-of-the art computer vision and language modelling research and then, used these technologies to create value for our customers especially in the e-commerce platform. We expect active discussion on that how to apply many existing research works into the e-commerce industry.

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
