# Peer review of "Multi-label learning with the RNNs for Fashion Search"

_ICLR 2017 — rejected_

[Author Response · Taewan Kim · 19 Nov 2016]
**Revision for minor typographical error**

We fixed minor typographical error in author's name and Section. 4. etc.
Our policy restricts to reveal much more details about the internal dataset and results of the end-user satisfaction measure, however, we did our best to introduce how our idea is to be used for multi-label learning in an application to computer vision, especially e-commerce industry.

[Official Review · AnonReviewer1 · rating 3 · confidence 3 · 17 Dec 2016]
**Contribution not clear enough; concerns about data set itself**

The manuscript is a bit scattered and hard to follow. There is technical depth but the paper doesn't do a good job explaining what shortcoming the proposed methods are overcoming and what baselines they are outperforming. 

The writing could be improved. There are numerous grammatical errors.

The experiments in 3.1 are interesting, but you need to be clearer about the relationship of your ResCeption method to the state-of-the-art. The use of extensive footnotes on page 5 is a bit odd. "That is a competitive result" is vague. A footnote links to "

[Official Review · AnonReviewer2 · rating 3 · confidence 4 · 19 Dec 2016]
**Good practical visual search system but lack novelty**

This paper introduces a pratical large-scale visual search system for a fashion site. It uses RNN to recognize multi-label attributes and uses state-of-art faster RCNN to extract features inside those region-of-interest (ROI). The technical contribution of this paper is not clear. Most of the approaches used are standard state-of-art methods and there are not a lot of novelties in applying those methods. For multi-label recognition task, there are other available methods, e.g. using binary models, changing cross-entropy loss function, etc. There aren't any comparison between the RNN method and other simple baselines. The order of the sequential RNN prediction is not clear either. It seems that the attributes form a tree hierarchy and that is used as the order of sequence.

The paper is not well written either. Most results are reported in the internal dataset and the authors won't release the dataset.

[Official Review · AnonReviewer3 · rating 4 · confidence 4 · 20 Dec 2016]
**interesting exploration but several major concerns**

The paper presents a large-scale visual search system for finding product images given a fashion item. The exploration is interesting and the paper does a nice job of discussing the challenges of operating in this domain. The proposed approach addresses several of the challenges. 

However, there are several concerns.

1) The main concern is that there are no comparisons or even mentions of the work done by Tamara Berg’s group on fashion recognition and fashion attributes, e.g., 
-  “Automatic Attribute Discovery and Characterization from Noisy Web Data” ECCV 2010 
- “Where to Buy It: Matching Street Clothing Photos in Online Shops” ICCV 2015,
- “Retrieving Similar Styles to Parse Clothing, TPAMI 2014,
etc
It is difficult to show the contribution and novelty of this work without discussing and comparing with this extensive prior art.

2) There are not enough details about the attribute dataset and the collection process. What is the source of the images? Are these clean product images or real-world images? How is the annotation done? What instructions are the annotators given? What annotations are being collected? I understand data statistics for example may be proprietary, but these kinds of qualitative details are important to understand the contributions of the paper. How can others compare to this work?

3) There are some missing baselines. How do the results in Table 2 compare to simpler methods, e.g., the BM or CM methods described in the text?

While the paper presents an interesting exploration, all these concerns would need to be addressed before the paper can be ready for publication.

[Final Decision · Program Chairs · 06 Feb 2017]
**ICLR committee final decision**

Three knowledgable reviewers recommend rejection. The main concern is missing related work on fashion product search, and thus also baselines. The authors did not post a rebuttal to address the concerns. The AC agrees with the reviewers' recommendation.